# A High-Quality Genome Assembly of Striped Catfish (*Pangasianodon hypophthalmus*) Based on Highly Accurate Long-Read HiFi Sequencing Data

**DOI:** 10.3390/genes13050923

**Published:** 2022-05-22

**Authors:** Dao Minh Hai, Duong Thuy Yen, Pham Thanh Liem, Bui Minh Tam, Do Thi Thanh Huong, Bui Thi Bich Hang, Dang Quang Hieu, Mutien-Marie Garigliany, Wouter Coppieters, Patrick Kestemont, Nguyen Thanh Phuong, Frédéric Farnir

**Affiliations:** 1FARAH/Sustainable Animal Production, Faculty of Veterinary Medicine, University of Liege (B43), 4000 Liege, Belgium; dmhai@ctu.edu.vn; 2College of Aquaculture and Fisheries, Can Tho University, Can Tho 900000, Vietnam; thuyyen@ctu.edu.vn (D.T.Y.); ptliem@ctu.edu.vn (P.T.L.); bmtam@ctu.edu.vn (B.M.T.); dtthuong@ctu.edu.vn (D.T.T.H.); btbhang@ctu.edu.vn (B.T.B.H.); quanghieudang.87@gmail.com (D.Q.H.); ntphuong@ctu.edu.vn (N.T.P.); 3FARAH/Veterinary Public Health, Faculty of Veterinary Medicine, University of Liege (B43), 4000 Liege, Belgium; mmgarigliany@uliege.be; 4Genomics Platform, GIGA, University of Liege, 4000 Liege, Belgium; wouter.coppieters@uliege.be; 5Research Unit in Environmental and Evolutionary Biology, Institute of Life, Earth & Environnment, University of Namur, rue de Bruxelles 61, 5000 Namur, Belgium; patrick.kestemont@unamur.be

**Keywords:** striped catfish, chromosome-scale genome assembly, selective breeding, HiFi reads

## Abstract

The HiFi sequencing technology yields highly accurate long-read data with accuracies greater than 99.9% that can be used to improve results for complex applications such as genome assembly. Our study presents a high-quality chromosome-scale genome assembly of striped catfish (*Pangasianodon hypophthalmus*), a commercially important species cultured mainly in Vietnam, integrating HiFi reads and Hi-C data. A 788.4 Mb genome containing 381 scaffolds with an N50 length of 21.8 Mb has been obtained from HiFi reads. These scaffolds have been further ordered and clustered into 30 chromosome groups, ranging from 1.4 to 57.6 Mb, based on Hi-C data. The present updated assembly has a contig N50 of 14.7 Mb, representing a 245-fold and 4.2-fold improvement over the previous Illumina and Illumina-Nanopore-Hi-C based version, respectively. In addition, the proportion of repeat elements and BUSCO genes identified in our genome is remarkably higher than in the two previously released striped catfish genomes. These results highlight the power of using HiFi reads to assemble the highly repetitive regions and to improve the quality of genome assembly. The updated, high-quality genome assembled in this work will provide a valuable genomic resource for future population genetics, conservation biology and selective breeding studies of striped catfish.

## 1. Introduction

The striped catfish (*Pangasianodon hypophthalmus)* is a freshwater species that is widely cultured in the Mekong River delta, Vietnam [1,2]. In 2021, striped catfish farming in the Mekong delta produced 1.48 million tons, being cultured on a farming area of approximately 5400 ha, with an export value of 1.61 billion USD [3]. Catfish aquaculture is, however, facing several challenges, including the lack of genetically selected broodstocks to respond to the high mortality caused by diseases and increasing salinity intrusion in the culture area due to climate change [4,5,6]. Use of the genomic information in selection programs might contribute by supplying necessary tools to tackle these challenges in striped catfish aquaculture and thus ensure its sustainability and profitability.

Genomic information provides powerful tools to enhance the fundamental research and applications for genetic improvement programs across many aquaculture species [7,8]. The whole genome sequence is now available for several economically important fish species such as channel catfish (*Ictalurus punctatus*) [9], Atlantic salmon (*Salmo salar*) [10], Nile tilapia (*Oreochromis niloticus*) [11], rainbow trout (*Oncorhynchus myskiss*) [12], and Atlantic cod (*Gadus morhua*) [13]. Based on these reference genomes, genomic markers such as single nucleotide polymorphisms (SNPs) or microsatellites can be detected and used in parentage assignment tools [14], and linkage maps can be obtained, allowing for example QTL/GWAS (Quantitative Trait Loci/Genome-Wide Association Study) to be performed. These genomic tools can then be used in breeding programs, including marker-assisted selection (MAS), genomic selection (GS), and genome editing. For instance, genomic resources have been used for mapping QTL for feed conversion efficiency in crucian carp (*Carassius auratus*) [15], GWAS for growth rate, and MAS for disease resistance in Atlantic Salmon (*Salmo salar*) [16,17].

For striped catfish, two versions of genome assemblies have been reported to date, with one based only on Illumina short reads [18] and another one based on an hybrid approach combining Illumina short reads, Nanopore long reads, and Hi-C data [19]. Illumina reads are accurate and have a low cost per base, but their read length, typically less than 600 bases, is shorter than many repeat elements in the genomes [20,21]. Consequently, de novo assembly using short reads often fails to resolve a full genome due especially to difficulties in the majority of repetitive regions. In contrast, long read sequencing technologies (or third-generation sequencing) can generate reads longer than 10 kb that can span highly repetitive regions of genomes and bridge previously intractable gaps in assemblies to improve overall contiguity [22,23]. The main drawback of long reads is a lower accuracy (an average accuracy of 90% for Nanopore and PacBio Sequel) [24] compared to Illumina short reads (99.9%) [25]. As a result, the draft genome produced only by long reads was more contiguous but also contained more errors [26,27]. A combination of the advantages of the two different sequencing technologies in so-called hybrid approaches can overcome their drawbacks [28,29]. In the hybrid approach, long-read assemblies are polished with Illumina short reads to improve accuracy [30,31]. On the negative side, most polishing tools may be unable to fix errors in repetitive regions that have ambiguous short-read mappings [26,32], and combining sequencing technologies may lead to increased costs and to more complexity in the projects.

As an alternative approach to eliminating dependencies on short-read data polishing, PacBio recently introduced high-fidelity reads (HiFi reads), with an average length of 13.5 kb and a substantial increase in the base accuracy compared to previous long reads (greater than 99.9%) [32,33]. The highly accurate long-read HiFi sequencing data have been proven to significantly improve the continuity and completeness of genome assembly in many studies [34,35]. Several new assemblers have been developed to fully exploit HiFi reads that do not require a final polishing phase [36,37]. It has been reported that the assembly exclusively based on long, highly accurate PacBio HiFi reads outperforms Illumina-Nanopore hybrid and Nanopore assembly. In addition, de novo assemblers using HiFi reads require lower amounts of data compared to other strategies. Using HiFi data, a 16× genome coverage is sufficient in order to produce high-quality assemblies [38]. For the human genome, the high accuracy of PacBio HiFi sequence data has improved variant discovery, reduced the assembly time, and provided access to even more complex regions of repetitive DNA [33,39,40]. With HiFi reads, more than 50% of the regions previously inaccessible with Illumina short-read sequence data in GRCh37 human reference genome are now accessible [39]. For most aquatic species, due to a high content of repetitive DNA sequences in their genomes, the assembly of the genome was extremely complicated [41]. For this reason, the HiFi sequencing technology may be a better choice to assemble the high-quality genome of striped catfish. Recent studies demonstrated that repetitive sequences play an important role in regulation of gene expression [42]. Consequently, a high-quality genome with a higher proportion of repetitive regions identified will offer more opportunities for a comprehensive understanding of the biology, especially in aquatic species [7]. Moreover, by using this improved reference genome, a larger number of DNA markers (e.g., SNPs) with high quality can be identified [41,43,44]. This Appendix A will facilitate MAS and GS to accelerate genetic improvement in selective breeding programs [7,45]. It has been reported that using a high-quality reference genome can bring several benefits for downstream genetic applications such as reduction in time and funds needed to implement MAS and GS and achievement of higher genetic gains in the breeding programs [7,41,46]. For example, a study on the blueberry (*Vaccinium corymbosum*), using a high-quality reference genome resulted in a higher precision about the location, number, and gene action of QTLs, which improved the understanding of the genetic architecture of the traits through GWAS analyses and the chances of uncovering the molecular mechanisms underpinning the trait variation [46].

In the present study, an improved (chromosome-level) assembly was developed for striped catfish by using PacBio HiFi long reads. The assembly was then scaffolded to chromosome level using Hi-C data. This improved reference assembly represents a step towards improving our understanding of fundamental biological and evolutionary questions and towards the genetic improvement of important aquaculture production traits via genomic-assisted breeding of striped catfish.

## 2. Materials and Methods

### 2.1. Sample Collection and DNA Extraction

This study was carried out using striped catfish collected from a hatchery in the Mekong delta, Vietnam (8°33′–10°55′ N; 104°30′–106°50′ E). High-quality genomic DNA was extracted from the blood of a mature, female, striped catfish with a weight of 5.5 kg (recently, a new study [47] suggested that striped catfish has a XY sex determination system) using NucleoBond HMW DNA kit (Macherey-Nagel, Düren, Germany) for PacBio sequencing and the Maxwell^®^ 16 Blood DNA Purification Kit (Promega, Madison, WI, USA) for Illumina sequencing. DNA quality and quantity were evaluated using electrophoresis on a 1% agarose gel and using Quant-iT™ PicoGreen^®^ dsDNA Reagent and Kits (Thermo Fisher Scientific, Waltham, MA, USA). Then, the DNA was sequenced using two libraries, with short and long reads.

### 2.2. Library Construction and DNA Sequence

The first library was generated to obtain long PacBio HiFi reads. The HiFi sequencing library was prepared using SMRTbell Express Template Prep Kit 2.0 (Pacific Biosciences, Menlo Park, CA, USA). The library was further size-selected using BluePippin System from SAGE science. The library was sequenced on one 8 M SMRT cell on Sequel II instrument using Sequel II Binding kit 2.0 and Sequencing chemistry v2.0. The sequencing produced 3,902,121 polymerase reads (~270.8 Gb of raw data) with an average polymerase read length of 69.4 kb and an average insert length of 16.4 kb. Circular Consensus Sequences (CCS) were generated using CCS pipeline (SMRT Link v9.0.0.92188) (PacBio, Menlo Park, CA, USA) with the following settings: minimum number of passes: 1, minimum predicted accuracy: 0.9. CCS reads with at least 99% accuracy are called HiFi reads. This sequencing was performed by the Norwegian Sequencing Centre (www.sequencing.uio.no) (accessed on 1 October 2020) from the University of Oslo, Norway.

In addition to PacBio reads, we employed Illumina paired-end sequencing by synthesis technology to sequence the striped catfish genome using the whole genome shotgun approach. Paired-end libraries with 350 and 550 bp fragments were constructed using a TruSeq DNA PCR-Free Kit (Illumina) according to manufacturer protocols. The libraries were sequenced using Illumina NextSeq 500 sequencing platform with Illumina protocols for whole-genome shotgun sequencing with paired end reads (2 × 150 bp). This work was performed by the Interdisciplinary Center for Biomedical Research (GIGA) at the University of Liege, Belgium.

### 2.3. Sequence Data Processing and Genome Assembly

Raw genomic library data generated by Illumina were quality checked using FastQC v.0.11.9 [48]. We performed data filtering and trimmed adapter sequences, low-quality reads and PCR duplication using Trimmomatic v 0.39 [49] with parameters ILLUMINACLIP: TruSeq3-PE.fa:2:30:20 LEADING:20 TRAILING:20 SLIDINGWINDOW:5:20 MINLEN:50.

The initial characterization of the striped catfish genome was estimated through *k-mer* (k = 21 in this study) analysis of the HiFi data by Jellyfish v 2.3.0 [50]. The histograms were uploaded to GenomeScope version 2.0 [51] for estimation of genome size, level of duplication, and heterozygosity.

The HiFi reads were assembled into contigs using Hifiasm-0.16.1 [52] using default parameter values. To scaffold genome, the cleaned Hi-C reads (with a total of 90 Gb) from a previous study on striped catfish of Gao et al. [19] were mapped to the primary assembly by BWA version 0.7.17 [53] to generate a BAM file that was subsequently converted to a BED file. Then, SALSA version 2.3 [54] used this BED file, which contained the mapping information of Hi-C reads on the assembly, to scaffold primary assembly.

To build chromosome-level assembly scaffolds, the cleaned Hi-C reads were first mapped to the assembled genome using BWA version 0.7.17 [53]. Scaffolds were then clustered, ordered, and oriented using Lachesis [55], with the parameter set “CLUSTER_N = 30, CLUSTER_MIN_RE_SITES = 25, CLUSTER_MAX_LINK_DENSITY = 2, CLUSTER_NONINFORMATIVE_RATIO = 3”. Figure 1 shows the assembly and annotation workflow.

### 2.4. Genome Quality Assessment

The quality of our final genome assembly was evaluated in terms of three data sets:BUSCO Actinopterygii Genes—Quality assessment was conducted using BUSCO software version 5.2.2 [56] with default parameters by searching the genome against 3640 single-copy orthologues from actinopterygii_odb10 database (https://busco-data.ezlab.org/v5/data/lineages/) (accessed on 10 November 2021);Illumina short DNA reads—To assess the accuracy of our genome assembly, we aligned the Illumina short reads to the assembly using BWA version 0.7.17 [53] to evaluate the mapping and covering rate;Comparison to other assemblies—Contig N50 values were calculated for comparison with those of previous studies on striped catfish [18,19] and other Siluriformes species by QUAST version 5.0.2 [57]. In addition, we also mapped our newly assembled genome against the previously published one: Genbank accession no. GCA_003671635.1 [18] and GCA_016801045.1 [19] to identify the gaps using MUMMER version 3.23 [58].

### 2.5. Identification of Repetitive Elements and Simple Sequence Repeat Markers

Prediction of repeat elements was based on de novo and homology methods. In the homology approach, we searched the genome for repetitive DNA elements using RepeatMasker version 4.1.2 [59] with parameters “-species *siluroidei*” based on the known repeats library (Dfam 3.5 database, https://www.dfam.org/releases/Dfam_3.5/families/) (accessed on 15 November 2021). To identify repetitive element de novo, RepeatModeler version 2.0.2 [60] was primarily used to generate a repeats library. Then, this library was aligned to the assembled genome with RepeatMasker version 4.1.2 [59]. In addition, the final assembled scaffolds were analyzed to identify simple sequence repeats (SSRs) such as di-, tri-, tetra-, pentra-, and hexa-nucleotide repeats using MISA version 2.1 [61].

## 3. Results

### 3.1. DNA Sequencing

For HiFi data, a total of 17.28 Gb (Q ≥ 20) were generated, corresponding to 1,168,528 reads with an average read length of 14,791 bp and a median of read quality of Q = 30. The distribution of read lengths and read qualities is presented in Appendix A. The data obtained from the two paired-end libraries of 350 bp and 550 bp insert sizes reached 51.2 Gb and 49.5 Gb, respectively (Table 1). A *k-mer* analysis predicted that the striped catfish had an estimated genome size of 713.9 Mb, the heterozygosity rate of the genome was 0.56%, and the repeats content of the genome was 17.5% (Appendix A).

### 3.2. Genome Assembly

The initial assembly of the striped catfish genome was based on HiFi reads only using Hifiasm de novo assembler, which resulted in a genome size of 788.1 Mb with 845 contigs and a contig N50 of 14.7 Mb. Next, the contigs were scaffolded using SALSA with Hi-C data, which generated a genome size of 788.4 Mb with 381 scaffolds and scaffold N50 of 21.8 Mb (Table 2). We also found that the average GC content in the striped catfish (38.96%) was similar to those in other species, which ranged from 33.62% to 42.01% [62]. Furthermore, 315 scaffolds (99.3% of genome) were successfully clustered into 30 chromosome groups. Finally, we obtained a high-quality, chromosomal-level genome with a total size of 783.1 Mb (Table 3).

### 3.3. Genome Quality Evaluation

BUSCO analysis of 3640 actinopterygii single-copy orthologues revealed that the proportions of complete (C), complete and single-copy (S), complete and duplicated (D), fragmented (F), and missing (M) genes were 96.0%, 93.5%, 2.5%, 1.0% and 3.0%, respectively (Table 4); 3497 (96.0%) core genes were completely identified in our assembly genome, suggesting a high completeness of genome assembly.

Using Illumina short reads, approximately 100% of these reads were mapped to our genome assembly, and these reads covered 93.5% of the total assembly.

For assessment using other assemblies, we compared our genome assembly to the two previous versions of the striped catfish genome as well as to those from other *Siluriformes* species. The results showed that our genome assembly had a similar size to those of other *Siluriformes* species (Table 5), and the contig N50 of the present genome assembly is significantly higher than those from the other previously published ones, including bighead catfish (GCA_011419295.1), walking catfish (GCA_003987875.1), and channel catfish (GCA_004006655.2) (Table 5). Moreover, the contig N50 value of our version of the genome (14.7 Mb) was 245-fold greater than the previous version of Kim et al. [18] based only on Illumina reads (0.06 Mb) and 4.2-fold greater than the version of Gao et al. [19] based on Illumina reads, Nanopore long reads, and Hi-C reads (3.5 Mb) (Table 6). In addition, the results from aligning our genome against the two previous versions indicated that 29,987 and 24,587 gaps were filled with respect to the versions of Kim et al. [18] and Gao et al. [19], respectively. These gaps corresponded to 72.7 Mb and 45.8 Mb of sequence in our new assembly, respectively.

### 3.4. Repetitive Genome Elements and SSR Markers

The results from an analysis of repeated elements showed that 308.36 Mb (39.11%) and 331.92 (42.10%) of the striped catfish genome consisted of repeat sequences when estimated from the de novo and homology approaches, respectively. For the de novo approach, approximately 57.2 Mb (7.25%) of class I retrotransposons were identified (long interspersed nuclear elements (LINEs), 3.87%; short interspersed nuclear elements (SINEs), 0.39%; total long terminal repeat elements, 2.99%). In addition, 101.79 Mb (12.91%) class II DNA transposons and 112.36 Mb (14.25%) unclassified elements were identified (Appendix A).

A total of 960,574 SSRs were obtained from the striped catfish genome assembly. The greatest fraction of SSRs were dinucleotides (53.2%), followed by mononucleotides (32.8%), trinucleotides (7.7%), tetranucleotides (5.6%), pentanucleotides (0.5%), and hexanucleotides (0.2%). Among these SSRs, T (49.2%) and A (48.6%) accounted for 97.8% of the total mononucleotide repeats. AC (17.3%), TG (15.9%), GT (15.7%), CA (14.3%), and TC (8.0%) accounted for 71.1% of the dinucleotide repeats, whereas AAT (11.3%), TTA (10.1%), ATT (9.9%), TAA (9.3%), TAT (7.7%), and ATA (7.4%) accounted for 55.7% of trinucleotide repeats (Appendix A).

## 4. Discussion

Genome assembly is the process of reconstructing a genome from randomly sampled sequence fragments in which overlapped sequence fragments are referred to as contigs (in the ideal case, one contig per chromosome) [33,63]. The quality of a genome assembly can be often evaluated using metrics related to contigs (contiguity), such as N50, and the ability to complete the whole structure of the genome (completeness), such as BUSCO score [64,65]. In the present study, a high-quality genome was assembled for striped catfish. This genome included 30 chromosomes, which is consistent with previously reported chromosome analysis in this species [66]. The contigs N50 value of the present genome assembly is significantly higher than in the previous versions (Table 6). In addition, the BUSCO value of the genome obtained in our study (96.0%) was slightly superior to that in the studies of Gao et al. [19] and Kim et al. [18] with 93.3% and 92.3%, respectively. Our findings were in agreement with previous works, where BUSCO scores and contig N50 of genome assembly were increased when using PacBio HiFi reads [67,68,69]. Moreover, we also obtained high mapping and coverage rates when we mapped the Illumina short reads on the obtained genome assembly. Together, these results indicated that our new genome assembly was more contiguous and complete than the previously published ones.

The long-read sequencing technologies can generate very long reads that make them invaluable for resolving complex repeat regions that cannot be assembled using shorter reads [70]. Among these technologies, the HiFi sequencing with both long and highly accurate reads have made it possible to overcome the assembly of highly repetitive regions [35]. In this study, the highly accurate long-read HiFi sequencing data used to generate the genome assembly have led to remarkable improvements, especially in finding new repetitive sequences. A higher number of repetitive elements (39.1%) were identified in our assembly compared to the previous genome assemblies of Gao et al. [19] with 36.9% and Kim et al. [18] with 33.8%. Our results are in accordance with prior work stating that a genome assembly using HiFi data has significantly improved continuity and accuracy in many complex regions of the genome and that the HiFi sequence data assembled an additional 10% of duplicated regions compared to one assembled using continuous long-reads (CLRs), which are currently the most common PacBio data type with lower accuracy (85–92%) [34]. In addition, our assembly also had a higher proportion of repeat sequences than Murray cod (*Maccullochella peelii*) with 23.38% [71] but lower than zebrafish [72] with 52.0%. Moreover, we identified 72.7 Mb and 45.8 Mb of new sequences that were unassembled in previous genome of Kim et al. [18] and Gao et al. [19], respectively. Combining this information with the increase in repetitive regions in our new assembly suggests that the new identified sequences might be located in repetitive regions of the genome. Taken together, these results are indicative of the high quality and completeness of the present genome assembly and its potential as an alternative to the previous versions of the striped catfish genome.

An improvement in the quality of a reference genome may often lead to novel discoveries. For example, improvements to rainbow trout (*Oncorhynchus mykiss*) assembly identified novel structure variants between two separate lineages and featured a scaffolded sex-determination gene (sdY) in the Y chromosome sequence [73]. The drastic improvements in the present genome could also lead to such new discoveries. It will also contribute to the acceleration of marker-assisted selection and functional genomic studies in striped catfish. Moreover, it could also facilitate more accurate marker ordering and fine mapping for QTL and GWAS.

## 5. Conclusions

In this study, we generated a high-quality, chromosome-level, reference genome assembly for striped catfish based on HiFi and Hi-C sequencing data. Both the sequence continuity (contig N50) and genome completeness (BUSCO score) were remarkably higher than those of the previously released striped catfish genomes, implicating the advantage of HiFi reads on de novo genome assembly. Moving forward, the reference genome presented here will serve as a springboard for further studies on the molecular evolution, comparative genomics, genotypic and phenotypic variation, genome structure, and genetic studies related to selective breeding on striped catfish.

## Figures and Tables

**Figure 1 genes-13-00923-f001:**
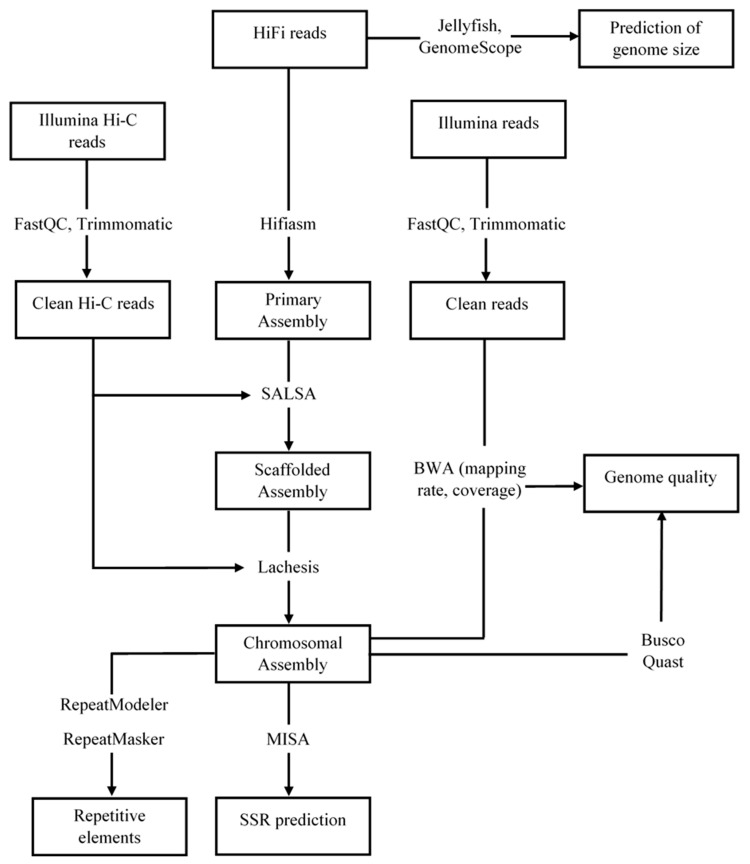
Detailed workflow for de novo whole-genome assembly and annotation.

**Table 1 genes-13-00923-t001:** Summary of sequencing data for striped catfish genome assembly.

Libraries	Insert Size (bp)	Total Data (Gb)	Read Length (bp)	Sequence Coverage (×) *
Illumina reads	350	51.2	150	71.8
Illumina reads	550	49.5	150	69.3
PacBio (HiFi) reads	16,400	17.28	14,791	24.2
Total		118.0		165.3

* The coverage was calculated according to estimated genome size of 713,911,345 bp.

**Table 2 genes-13-00923-t002:** Summary of the final striped catfish genome assembly.

Category	Contig		Scaffold	
	Length (bp)	Number	Length (bp)	Number
Total	788,121,403	845	788,355,903	381
Largest	30,145,618	NA	35,439,358	NA
N50	14,675,983	20	21,837,136	15
N60	11,176,154	26	19,579,817	19
N70	6,525,798	35	17,728,427	24
N80	2,330,994	55	14,533,765	28
N90	459,384	139	1,263,789	54

N/A: not applicable.

**Table 3 genes-13-00923-t003:** Statistics of chromosomal level assembly of striped catfish.

Chr ID	Length (bp)	Chr ID	Length (bp)	Chr ID	Length (bp)
Chr1	57,614,776	Chr11	30,308,490	Chr21	20,241,537
Chr2	57,569,486	Chr12	30,160,703	Chr22	20,107,636
Chr3	47,895,633	Chr13	27,700,161	Chr23	19,020,310
Chr4	40,963,295	Chr14	26,554,853	Chr24	18,006,869
Chr5	37,055,990	Chr15	26,362,405	Chr25	16,472,023
Chr6	33,881,059	Chr16	25,536,520	Chr26	12,481,112
Chr7	33,777,686	Chr17	25,407,408	Chr27	6,262,317
Chr8	33,526,538	Chr18	22,657,484	Chr28	2,381,093
Chr9	32,744,715	Chr19	22,587,380	Chr29	1,580,235
Chr10	32,245,061	Chr20	20,661,286	Chr30	1,373,485
Total chromosome-level length			783,137,546		
Total length			788,355,903		
Chromosome length/total length			99.3%		

**Table 4 genes-13-00923-t004:** Genome assessment based on BUSCO annotations.

Index	Number
Complete BUSCOs (C)	3497
Complete and single-copy BUSCOs (S)	3405
Complete and duplicated BUSCOs (D)	92
Fragmented BUSCOs (F)	35
Missing BUSCOs (M)	108
Total BUSCO groups searched (n)	3640
C: 96.0% [S: 93.5%, D: 2.5%], F: 1.0%, M: 3.0%, n: 3640	

**Table 5 genes-13-00923-t005:** Comparison of the genome assemblies of various *Siluriformes* species.

Species	Genome Size (Mb)	Number of Contig	Number
*Pangasianodon hypophthalmus* (from the present study)	788.4	845	14.7
*Clarias macrocephalus* (GCA_011419295.1)	883.3	44,869	0.05
*Clarias batrachus* (GCA_003987875.1)	821.7	78,047	0.02
*Ictalurus punctatus* (GCA_004006655.2)	1002.3	5816	2.7
*Ageneiosus marmoratus* (GCA_003347165.1)	1030	169,048	0.007
*Ompok bimaculatus* (GCA_009108245.1)	718.1	27,068	0.08
*Bagarius yarrelli* (GCA_005784505.1)	570.8	928	1.9
*Tachysurus fulvidraco* (GCA_003724035.1)	713.8	2402	1.0

**Table 6 genes-13-00923-t006:** Comparison of quality metrics of this study and the previous striped catfish genome assemblies.

Genomic Feature	This Study	Gao et al. [19]	Kim et al. [18]
The size of genome (Mb)	788.4	742.6	715.7
Number of contigs	845	821	23,340
Contig N50 (Mb)	14.7	3.5	0.06
Longest contig (Mb)	30.1	16.1	0.5
GC content (%)	38.9	38.9	38.7
Repetitive regions (%)	39.1	36.9	33.8
Complete BUSCOs (C) (%)	96.0	93.3	92.3

## Data Availability

The raw sequencing data presented in this study are deposited in NCBI under BioProject PRJNA826868.

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
