# Peer review of "A High-Quality Genome Assembly of Striped Catfish (Pangasianodon hypophthalmus) Based on Highly Accurate Long-Read HiFi Sequencing Data"

_genes, 2022, doi:10.3390/genes13050923_

Round 1

Reviewer 1 Report

Hai et al. made a substantial effort to develop a highly contiguous genome assembly of a commercially important fish species by using a combination of complementary sequencing technologies. This is a nice piece of a technical report, employing the latest standard approach. Analytical approaches seem appropriate, and the genome assembly statistics met the authors’ expectations. I do not have much to add to the current version. I look forward to authors’ future studies by using this new genome assembly to explore the genetic basis of various commercially important traits and improve traits by marker-assisted selection.

Specific comments:

Line 67 “PacBio Seque I” -> Sequel

Line 57-75. This is a well-written and concise summary of the pros and cons of various sequencing technologies. I would suggest the following two points be added to this paragraph. First, please provide some summary statistics about the available two genome assemblies of the striped catfish (ref 18 and 19) in order to characterize what is the most important information that is missing in the reference genome reported in ref 19. By doing so, the authors can better motivate how this study filled the missing information relative to the previous versions of the genome assemblies. Second, please describe why we need more “complete” or better reference genomes. I fully agree that a good reference genome is a “necessary tool tackle these challenges in striped 41 catfish aquaculture” (Line 41), but it is not clear why a better genome can help many of the analyses described in the paragraph (line 43-56), such as GWAS and MAS. For example, the authors suggest that PacBio HiFi (Line 77) can resolve some ambiguities in repetitive regions, but how can the better characterization of these regions help GWAS and MAS? Some of these points are indirectly touched on in the following paragraph (eg., line 87-91), but putting this in a more relevant context (for example, in this specific study species) would help readers understand what the authors are envisioning to achieve by having a high-quality reference genome of striped catfish.

Line 183, Please double-check the number “1,68,528”

Table 1: The bottom cell of the column “Insert size” should be empty, I believe.

Author Response

Thanks to the reviewer for his/her evaluation of our paper! We have tried to reply to all the questions. In the following documents, we have reported the questions in black and our answers in blue. We hope that you will consider our replies as satisfactory. Thanks again for considering our paper !

Sincerely,

The authors

Reviewer 2 Report

Genes-1712001-REVIEW

This paper applied a new sequencing technology- PacBio HiFi in striped catfish, which resolved a high quality genome assembly compared to Illumina-Nanopore hybrid and Nanopore assembly, as revealed in some aspects of contig N50 numbers, BUSCO value, and number of repetitive elements.

The striped catfish is a important freshwater species, widely cultured in Vietanm. Catfish aquaculture is facing several challenges, the high mortality caused by diseases and increasing salinity intrusion in the culture area due to climate change. Presently, there is a lack of genetically selected broodstocks of striped catfish, however, some genome assembly works developed quickly in recent years. How to answer/resolve the problem in catfish aquaculture ---- the higher mortality? How to fill/ minimize the gap between developing selected broodstocks/ strains (weak work) and its genome assembly (more progressed).

For striped catfish, two version of genome assemblies have been reported to data, with one based only on Illumina short reads [18], and another one based on an hybrid approach combining Illumina short reads, Nanopore long reads and Hi-C data [19]. 

Which shortages or problems existed in marker-assistance selection, or QTL, GWAS analysis using the above two genome assemblies?

To overcome these problems, what merit of the genome assembly by HiFi sequencing?

This improved reference assembly represents a step to wards improving our understanding of fundamental biological and evolutionary questions and towards the genetic improvement of important aquaculture production traits via genomic-assisted breeding of striped catfish

In result, which information/data in this updated, high quality genome assembly, are more helpful for understanding of fundamental biological and evolutionary questions, or genomic-assisted breeding program?

Line 101-102: “High quality genomic DNA was extracted from blood of a striped catfish.....” Give some basic information about the sample fish, for example, the size, sex, etc.

Author Response

(The authors gave the same response as above.)

Round 2

Reviewer 2 Report

no comment.

Author Response

Thanks for reviewing again our paper.

Since you had no comments for the authors, we assume that you found the scientific content satisfactory.

You noted that various points could be improved, but we have no additional information on your expectations. We have nevertheless implemented all editorial demands from the editor, we hope that you will consider these little changes as sufficient.

Sincerely,

The authors